# A New Approach to the Development of Geothermal Water Utilization in the Context of Identifying and Meeting the Social Needs of Local Communities: A Case Study from the Mogilno–Łódź Trough, Central Poland

Anna Wachowicz-Pyzik *, Anna Sowiżdżał [ID], Tomasz Maćkowski [ID] and Michał Stefaniuk [ID]

Faculty of Geology, Geophysics and Environmental Protection, Department of Fossil Fuels, AGH University of Science and Technology, 30-059 Kraków, Poland; ansow@agh.edu.pl (A.S.); mackowsk@agh.edu.pl (T.M.); stefaniu@agh.edu.pl (M.S.)
* Correspondence: amwachow@agh.edu.pl

**Abstract:** For many years, geothermal energy has been successfully used for both energy as well as balneological, healing, and recreational purposes. It should be emphasized that, along with the great interest in geothermal investments, the development of other economic sectors (i.e., tourism, cosmetology, food production, and many other sectors related directly or indirectly to geothermal waters) are also noted in this paper. That kind of development is seen both in regions where centers using geothermal energy are created, as well as in their immediate vicinity. An important aspect of the use of geothermal energy is also its positive impact on the environment by reducing the emission of pollutants that could end up in the environment as a result of using conventional energy sources, namely coal or natural gas. Given the high level of air pollution in Poland, according to data from the European Environment Agency, 12 Polish cities are among the 20 most polluted cities in Europe (data for 2021–2022), and this aspect seems to be key for achieving sustainability while maintaining economic balance. In this article, a new approach to the development of geothermal water utilization in the context of identifying and meeting the social needs of local communities in the Mogilno–Łódź Trough region is described.

**Keywords:** geothermal water; geothermal energy; Poland; Mogilno–Łódź Trough

## 1. Introduction

The use of geothermal energy in Europe is projected to increase rapidly in the coming decades due to the many advantages that this type of energy has to offer. Geothermal resources are generally abundant, ubiquitous, versatile, low-carbon, and non-intermittent [1]. However, real possibilities for the effective management of geothermal resources do not exist everywhere. The economic potential for geothermal electricity generation in 2050 exceeds 500 GW in Europe [2]. According to the European Geothermal Energy Council [3], at the end of 2019, there were 130 operating geothermal installations in Europe, 36 projects under development, and 124 projects in the planning phase. It is predicted that the number of operating plants could double in the next 5–8 years. Europe is a leading global market for geothermal district heating and cooling for buildings, industrial services, and agriculture. In 2019, there was 5.5 GWth of installed geothermal district heating and cooling capacity in 25 European countries, corresponding to 327 systems.

In addition to favorable hydrogeothermal conditions, the impulse to increase the use of geothermal energy in recent years is undoubtedly the result of the directives imposed by the European Union (EU) on the member states. These directives are related to the necessity of reducing environmental pollution, as well as the search for alternative solutions that, in the future, may partially or completely supplant conventional energy carriers such as coal or natural gas.

Due to the presence of low-temperature hydrogeothermal resources, Poland is currently outside the group of countries producing electricity from geothermal resources, although an opportunity for growth in this sector would be to use the energy of deeply deposited hot dry rocks (petrogeothermal resources). Such projects are still only in the analytical phase [4]. On the other hand, there are real possibilities for managing geothermal resources for heating purposes, as evidenced by the temperature of geothermal waters (documented to range from 20 to over 90 °C) and high local capacity exceeding 300 m³/h [4], as well as existing geothermal plants. Currently, there are seven geothermal heating plants in Poland, six of which are located in the Polish Lowlands (Figure 1). These are as follows: Geotermia Pyrzyce (built in 1997), Geotermia Mszczonów (built in 1999), Geotermia Uniejów (built in 2000), Geotermia Stargard (currently G-Term Energy; built in 2005), Geothermal Heating Plant Poddębice (built in 2013), and Geotermia Toruń (built most recently, in 2022). The seventh geothermal heating plant (Geotermia Podhalańska) is the only one located in the south of the country, in the Podhale region (Inner Carpathians), and it is the oldest (1993) and largest geothermal heating plant in Poland [5].

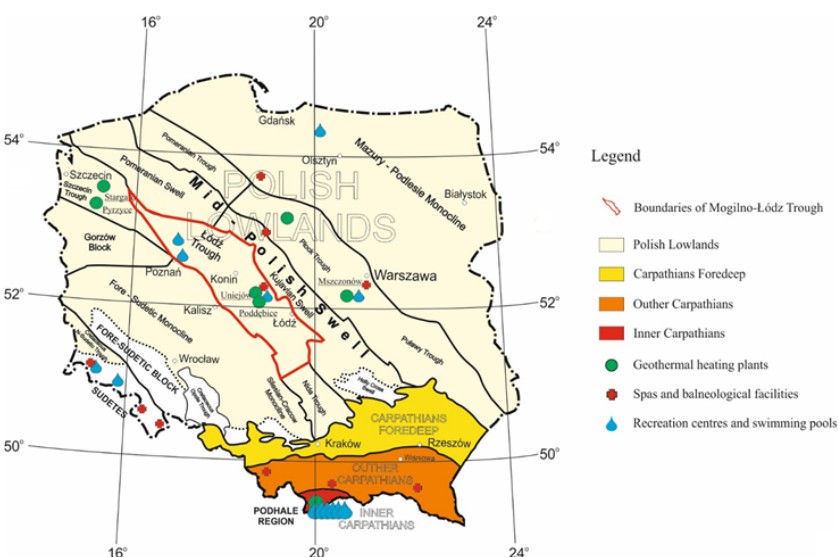

**Figure 1.** Location of heating plants and centers using geothermal water in Poland (adapted from Ref. [5]).

In addition to heating plants that use groundwater directly, recreational, balneological, and therapeutic centers utilizing thermal waters have also played an important role in recent years. Currently, there are 10 health resorts in Poland (Figure 1), of which the newly opened facilities include the health resort in Marusza (opened in 2009), the Rabka Zdrój health resort (opened in 2011), and the Uniejów health resort (which has been in operation since 2012). These facilities use groundwater in a temperature range of approximately 18 to 60 °C [5,6].

Recreation centers and bathing areas are equally important for the further development of geothermal energy use. Currently, a total of 14 such facilities have been opened in Poland (Figure 1), half of which are located in Podhale, and the other half in the Polish Lowlands. Due to the great interest of the tourist industry in recreation centers that utilize geothermal waters, more and more often in the vicinity of facilities using geothermal energy, hotel facilities are being created. Also, high-class SPA centers which also use groundwater to provide a wide range of services, from care to therapeutic treatments (e.g., the Medical Spa Hotel Lawendowe Termy in Uniejów or Termy Bania in Białka Tatrzańska), are becoming more and more popular. Thermal waters are commonly used to treat diseases such as cardiovascular diseases, rheumatoid diseases, and gastric problems. They are also used for injury rehabilitation and relaxation treatments (e.g., pearl baths).

Furthermore, thermal waters are used, among others, for Atlantic salmon farming in Janów near Trzęsacz (launched in 2015) to heat the grasslands of playing fields or walking paths, as well as in food and cosmetic processing, in the cases of Pyrzyce and Uniejów [5]. In Poland, a noticeable increase in demand for recreational facilities based on geothermal waters and a great interest in newly built facilities [7–9] have been observed.

Despite the various uses of geothermal energy in Poland, for many years, this sector has ranked in last place compared with other sectors in terms of it use of Renewable Energy Sources (RES), according to the Polish Central Statistical Office (Figure 2). Taking into account the sustainable development of renewable energy sources in Poland, a new approach for the utilization of geothermal waters should be implemented.

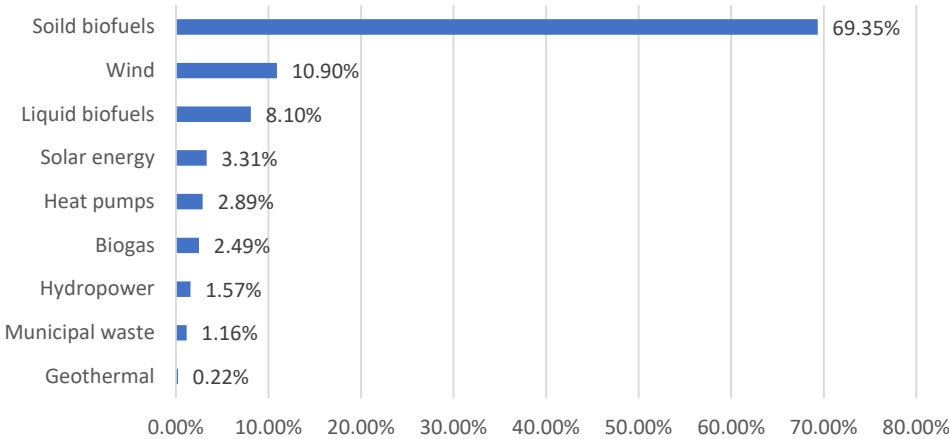

**Figure 2.** The production of energy from renewable sources by carriers in Poland in 2021 (based on statistics from the Central Statistical Office).

## 2. Methodology

In accordance with the conducted research and analyses [4,8,10,11], taking into account the current level of groundwater utilization, the Moglino–Łódź Trough region was characterized. Our analysis also covered the social aspects of the use of geothermal waters, including the different ways of using the water and the social benefits derived from its use. In this paper, we propose the extension of the benefits associated with the use of geothermal energy by increasing security regarding the stability of supplies and the availability of energy sources. The multifaceted applications of geothermal waters in the area of the Mogilno–Łódź Trough in Geotermia Uniejów and Poddębice are presented in Section 3.1. Based on the results of a study carried out as part of a PhD thesis [12], a new prospective region for the use of geothermal energy utilization (Malanów region) is also presented in this paper. Based on the data from the Central Statistical Office for the years 2017–2021, we predicted the production of geothermal energy for the commerce and public service sectors in Poland until 2030.

## 3. The Geothermal Water of the Mogilno–Łódz Trough

A particularly affluent region in Poland in terms of the possibility of using geothermal energy is the Mogilno–Łódź Trough, especially its southwestern area, where, in the 1970s, groundwater with a temperature of 68 °C was drilled, thus confirming the presence of geothermal waters within the Lower Cretaceous. Subsequent wells drilled in the 1990s, and waters with a capacity of 90 m$^3$/h and a temperature of approximately 70 °C have been documented. The deposits of the Lower Cretaceous reservoir layer are sandstones located at a depth of about 1.9–2.0 km. In the area of the Mogilno–Łódź Trough, the depth of the occurrence of fresh (ordinary) waters reaches approximately 1700 m. However, at a relatively short distances from these waters, there are waters (brines) with high mineralization, up to several dozen g/dm$^3$. This is associated with the operational decrease in

primary formation pressures and may result in the ascent of saline waters from the deeper parts of the reservoir or from the lower aquifers (e.g., the Jurassic reservoir) [10].

Currently, there are two geothermal heating plants, located in Uniejów and Poddębice, in the area of the Mogilno–Łódź Trough. In both cases, the obtained geothermal energy comes from the Lower Cretaceous reservoir. In Uniejów, the Lower Cretaceous deposits are composed of sandstone and sand–carbonate complexes located at a depth of 3000 m below sea level at groundwater temperatures ranging from 110 to 115 °C [10,11,13]. In 1978, by the Uniejów IGH-1 well located in the village of Ostrowsko, a team led by Zbigniew Płochniewski initiated the geological and deposit recognition of the Lower Cretaceous thermal waters, which are currently used in the Uniejów geothermal heating plant [13–15]. After drilling the Uniejów IGH-1 well in 1990–1991, further wells were created, Uniejów PIG/AGH-1 and Uniejów PIG/AGH-2 [16], according to a project developed by the AGH University of Science and Technology in Kraków under the supervision of Professor Górecki. Initially, the project assumed the creation of three wells, which, together with the previously made Uniejów IGH-1 well, were to form a double geothermal doublet. However, due to a lack of adequate financial resources, only two of the three planned wells were drilled [15]. These wells reach the Lower Cretaceous aquifer composed of quartz, light gray, and multi-grained sandstones and locally coarse-grained sandstones, which, in the bottom part, pass into fine-grained and very-fine-grained sandstones with mudstone intercalations [14]. The Lower Cretaceous deposits are characterized by an effective porosity coefficient of 18–20%, close to the value of the total porosity, which proves the good hydraulic connection of the pores, while the permeability values reach approximately 359.7 mD [14,15].

In addition to heat plants that use groundwater directly, recreational, balneological, and therapeutic centers utilizing thermal waters have also played an important role in recent years. Apart from two heat plants in the Mogilno–Łódź Trough, within its borders, there are also recreation centers in Poznań, Tarnów Podgórny, and for the coming years, further geothermal projects and investments are being planned for cities such as Konin, Turek, and Koło [4,5,16].

Particularly noteworthy is the project based in Konin on account of the Konin IG-1 well drilled in recent years. The construction of this well started in September 2014. Initially, the first Lower Cretaceous aquifer was drilled to a depth of 1620 m below sea level, obtaining a water temperature of 62 °C and mineralization of 35 g/L, with a capacity of 300–500 $m^3$/h [17]. Finally, the well was completed at a depth of 2660 m, thus drilling the second aquifer of the Lower Jurassic reservoir, obtaining a water temperature of 97.5 °C, with mineralization of 150 g/L. On the basis of physicochemical tests, it turned out that these are highly mineralized waters of the sodium chloride type. Large amounts of chloride, sodium, magnesium, and calcium ions, as well as microelements, make it possible to use the water for healing purposes. Taking into account the high water temperature obtained, the possibility of producing electricity has also been considered. If used for this purpose, the geothermal plant in Konin would become a pioneer in the use of groundwater for the generation of electricity [17].

### 3.1. The Multifaceted Applications of Geothermal Waters in the Area of the Uniejów

Uniejów, located in the Mogilno–Łódź Trough, hosts examples of the multifaceted applications of geothermal waters. The geothermal heating plant in Uniejów started operating in 2001 by exploiting groundwater from the Lower Cretaceous sandstones from a depth of 1982–2084 m below sea level [10]. The plant consists of three wells: one for production, (Uniejów PIG/AGH-2) and two for injection (Uniejów PIG/AGH-1 and a reconstructed well named Uniejów IGH-1). The system capacity is 120 $m^3$/h at a temperature of 68 °C, and mineralization ranges from 6.8 to 8.8 g/L. The Lower Cretaceous waters used in Uniejów are sodium chloride waters [18]. The geothermal system initially operated in the geothermal doublet, but since 2004, it has been working on the basis of tripled wells consisting of a production well and two absorbent wells [14]. Due to the decrease in the absorptive capacities of both injection wells [19], their operation was stopped (in 2008 in the

case of the Uniejów IGH-1 well and two years later for the Uniejów AGH/PIG-1 well) [20]. This decrease in absorbency was related to the phenomenon of clogging and technical errors both during the drilling of the well and its subsequent operation [18]. Currently, geothermal energy works with the use of a geothermal doublet thanks to the restoration of the PGI/AGH-1 well, while the Uniejów IGH-1 well is used as a backup well in case of the working wells experience any failures or undergo maintenance. The power of the heat plant in Uniejów reaches 7.4 MW, of which 3.2 MW comes from geothermal waters, and the remaining part is covered by two biomass boilers, which are the peak heat sources of the system.

The district heating network, powered by geothermal energy, covers 70% of the city's heat energy needs (Figure 3). The geothermal system in Uniejów is based on two boreholes (production and injection). Geotermia Uniejów is the first heating plant in Poland to use exclusively renewable energy sources, geothermal water and biomass. Currently, the total power of the heating plant is 7.4 MW, of which 1.8 MW comes from biomass combustion, and 3.2 MW comes from geothermal water energy. In addition, the system is equipped with oil boilers with a capacity of 2.4 MW, which serve as a reserve [21]. The use of geothermal water in Uniejów is based on so-called cascade energy reception, which centers around the fact that the water extracted to the earth's surface first transfers some of its thermal energy in heat exchangers to the district heating water. This helps avoid the risk that the minerals contained in the water will damage the installation and the heating network. Fresh water heated thanks to geothermal energy goes further to recipients, heating their rooms and domestic hot water. In turn, chilled water finds its applications in balneotherapy and recreation, and it is used for heating the turf of sports fields or walking paths. Finally, the geothermal water, which has already given up as much of its energy as possible, is pumped back below the earth's surface, where it heats up again and may be recovered in the future. In this way, the entire cycle is repeated, guaranteeing the renewability of the energy source.

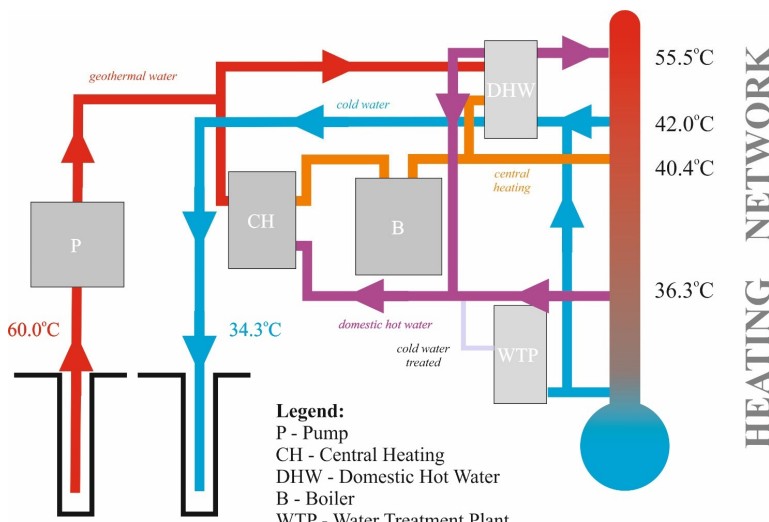

**Figure 3.** District heating system for the use of geothermal waters in Uniejów (adapted Ref. [21]).

The geothermal water resources in the Uniejów region have thermal and chemical parameters which support the wide economic use of the raw material. The parameters of geothermal water (Table 1), namely its temperature (68 °C), self-outflow capacity, and mineralization, have a beneficial effect on the exploitation process and the use of geothermal deposits [21]. Due to the content of iodides (0.42 mg/dm$^3$) and fluorides (0.7 mg/dm$^3$), thermal water can also be used in drinking cures in the case of iodine and fluoride deficiencies. A factor that especially interacts with bathing procedures in the thermal water of Uniejów is the lowland climate, which is especially beneficial for people with rheumatic diseases and respiratory diseases, and during convalescence after many diseases. Since 2012,

Uniejów has enjoyed the status of a health resort, which is related to the healing properties of geothermal waters as well as the climate. To obtain the title of a health resort, certificates and documentation confirming the existence of health-promoting natural resources in the town are necessary [22].

**Table 1.** The sensory and physicochemical properties of geothermal water from the PGI/AGH-2 well in Uniejów (Source: PIG-AGH-2 water well in Uniejów on 29 September 2008, No.HU-96/WL/AN/08, National Institute of Public Health, National Institute of Hygiene in Warsaw).

| | |
|---|---|
| Color [mg Pt] | 0 |
| Taste | Saline |
| Smell | Very weak oil |
| Water reaction [pH] | 7.20 (22 °C) |
| Temperature of water [°C] | 68 |
| Redox potential E [mV] | −51.9 (22 °C) |
| Conductivity [S/cm] | $12.06 \times 10^{-3}$ |
| Absorption $\lambda = 254$ [nm] | 0.059 |
| Absorption $\lambda = 436$ [nm] | 0.005 |

The water extracted in Uniejów has smoothing and moisturizing properties, and its optimal pH allows it to be used as a base for the production of cosmetics. The metasilicic acid contained in the water from the Uniejów thermal baths improves skin hydration, i.e., skin's elasticity and firmness. This ingredient causes a slight browning of the skin. The radon contained in the water enriches the skin's vital powers, while fluoride improves oral hygiene. Compounds of copper and iron contribute to the strengthening of the body's immunity. The line of cosmetic products derived from this water includes products such as face cream for daily use, face wash gel, micellar liquid, face mist, and shower gel (for body and hair washing) [23].

Uniejów is an excellent example of the use of geothermal waters not only for the purpose of obtaining thermal energy but also for the production of juices, soups, cosmetics, and preparations for animals, which translates into the simultaneous development of various economic sectors, as well as the creation of new jobs for the inhabitants of the town and the surrounding area.

*3.2. The Poddębice Geothermal Heating Plant*

The second heat plant located in the Mogilno–Łódź Trough is the Poddębice heat plant, where geothermal waters were documented in 2010. Currently, geothermal water is used for heating purposes, balneotherapy, recreation and, on a limited scale, also for consumption. An interesting case, from chemical and thermal points of view, is water from the Lower Cretaceous at great depth, which was made available through the Poddębice GT-2 well. A sampled interval is located at depths of 1962–2063 m. The geothermal water at the outflow of the well is characterized by having a temperature of ca. 71 °C and low mineralization of 432 mg/L, which is unusual for waters occurring in the surrounding area at similar depths. The diagram in Figure 4 shows the main water ions in this water. The further expansion of the system for the following years has planned, including the revitalization and development of the existing geothermal complex. Water from the Poddębice GT-2 well is of the Na–Ca–$HCO_3$ type, which distinguishes it from other waters from the area, where waters of the Na–Cl type prevail. Their mineralization ranges from ca. 2–9 to 21–74 g/L, which is much higher than the mineralization of the fresh water from the Poddębice GT-2 well. Moreover, water from the well also contains slight amounts of iron, iodine, and bromine ions [24]. Poddębice, following the example of the heat plant in Uniejów, is trying to leverage the hidden potential of groundwaters as widely as possible [5].

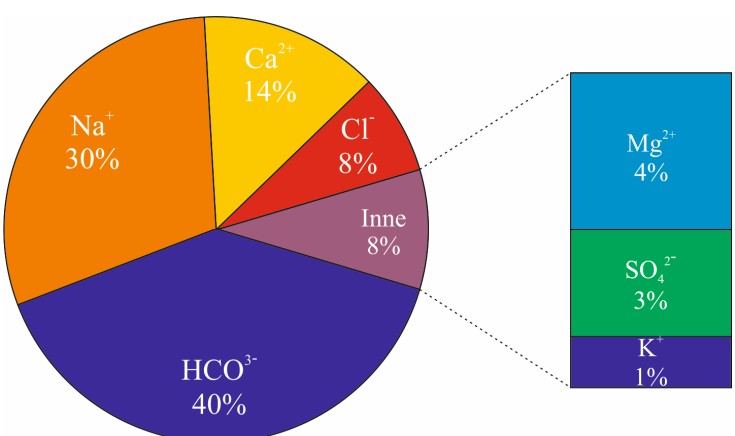

**Figure 4.** Main ion contents of the waters from the Poddębice GT-2 well. Values in meq/L [24].

*3.3. Opportunities for Geothermal Development in the Malanów Region*

Due to the great potential of the Mogilno–Łódź Trough, based on research and analyses carried out so far [24,25], another area that could be developed in a similar way in the near future is the Malanów region (Figure 5). This area, located in central–western Poland within the Wielkopolskie Voivodeship, is poor in mineral resources, mainly natural aggregates, sand, gravel, and peat, but they are not of significant importance for their economic use. Large tracts of forests and numerous peat bogs, swamps, and meadows have an influence on the landscape and promote the development of tourism. There is no centralized heating network in Malanów and its neighboring communes. The buildings are mostly heated by independent heating installations fueled by coal, fuel oil, or wood. Taking into account the development possibilities of regions using geothermal energy, as is the case of Uniejów or Poddębice, potential investments in the Malanów region could contribute to the development of both the Malanów commune itself and its neighboring communes.

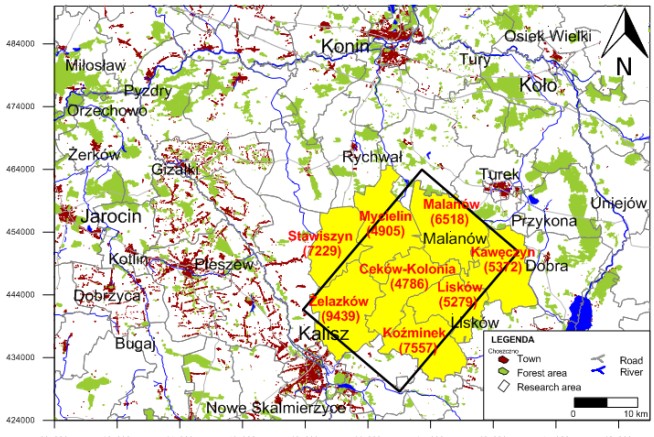

**Figure 5.** Location of communes in the Malanów region. The number of inhabitants is given in brackets (according to data from the Central Statistical Office).

A detailed analysis of the Malanów region was conducted as part of a PhD thesis [12]. Based on the data from deep boreholes and geophysical data including seismic reinterpretation data, a local static model (structural and parametric) was developed. Based on this model, it was possible to assess the most important geothermal parameters from the point of view of the possibility of using geothermal energy, i.e., silt content, porosity, or permeability. The results of static modeling confirmed that the analyzed area has good reservoir properties, related mainly to the Sinemurian deposits. Table 2 summarizes the mean values obtained during the parametric modeling for the three analyzed horizons of the Lower Jurassic.

**Table 2.** A summary of mean values of silt content, effective porosity, and permeability for successive horizons of the static model of the Malanów region [12].

| Horizon | Silt Content [-] | Effective Porosity [%] | Permeability [mD] |
| --- | --- | --- | --- |
| Toarcian | 0.45 | 6.5 | 44 |
| Pliensbachian | 0.32 | 10 | 120 |
| Sinemurian | 0.30 | 10 | 130 |

As part of the analysis, the value of the intake potential permeability was also estimated (Table 3). On the basis of the obtained results, the best parameters are characteristic of the following horizons: Sinemurian and Pliensbachian, whose maximum intake potential permeability was 120 m$^3$/h. In the case of the Pliensbachian, the capacity does not exceed 40 m$^3$/h in the majority of the area. Analyzing the obtained parameter results, the best areas for a possible geothermal investment in this region are the vicinity of the Malanów-1 well, where the depth of water intake does not exceed 2.500 m above sea level, with a water temperature of 86 °C and a maximum estimated permeability of 120 m$^3$/h.

**Table 3.** Comparison of potential well capacity values for the successive horizons of the static model of the Malanów region [12].

| | Potential Well Capacity Value [m$^3$/h] | |
| --- | --- | --- |
| Horizon | Min. | Max. |
| Toarcian | 5 | 80 |
| Pliensbachian | 10 | 120 |
| Sinemurian | 15 | 120 |

Given the temperature of the groundwater in the Malanów region, which is above 80 °C, it could be used in a variety of applications, ranging from fish farming to air conditioning and the cooling and heating of rooms. Taking into account the existing technical solutions, an alternative to increasing the temperature of the intake water is the use of heat pumps in the heating system, thanks to which it is possible to extend the scope of the use of the geothermal water with lower temperatures. Heat pumps, among others, are now successfully used in the heat plant in Pyrzyce and across the Podhale region. However, with the aim of achieving the optimal use of geothermal waters, it is recommended to use the so-called cascade systems based on the gradual cooling of the water after the heat is transferred to the network water (stage I) through successive cooling stages (e.g., stage II for heating swimming pools and stage III for heating the turf of a sports field, as is the case with the heating plant in Uniejów).

The chemical composition of the Lower Jurassic (Lias) waters, which are mostly of the Na–Cl type and contain some specific components (I, Br, and Fe), is also worth mentioning. The value of bromides and iodides is not enough to justify their use for industrial extraction, but they are sufficient for the production of salts added to fresh waters and for bathing or other therapeutic purposes. Such waters can also be used as a base for cosmetic formulas [23].

It should be emphasized that the key element in the utilization of geothermal waters is the potential recipients. In the case of the Malanów region, the closest potential recipient is the city of Kalisz (over 100.000 inhabitants). However, due to the large discrepancies in the hypothetical investments for the city, it is difficult to determine the profitability of such an undertaking. Still, it would be possible to supply heat to the communes in the immediate vicinity of the analyzed area.

## 4. Discussion—Social Aspects of the Use of Geothermal Waters

### 4.1. Ways of Using Geothermal Waters

The greatest advantage of using geothermal resources is the multifaceted nature of their uses. The history of their use is thousands of years long, and geothermal resources have also been a part of the past of many civilizations and nations [26]. In Japan, for instance, people used to settle near hot springs as early as 11.000 BC, and in mainland Asia, this practice dates back to as early as 5000 BC [27]. Geothermal waters have been widely used for years for their healing properties. They were also highly valued in Chinese medicine, which developed the principles of treatment with geothermal waters [28]. Thermal baths owe their development to a large extent to geothermal waters. They were first built by the Etruscans, and they were brought to perfection by the Romans who used Greek patterns [26]. In the Roman Empire, they were associated with a ritual, an art of rest and social and political life. Moreover, they played a significant role in shaping urban communities, trade, and economic ties [29].

The ways of using the energy accumulated in geothermal waters and vapors can be divided into two main groups [10]. The first is the generation of electricity using geothermal vapors; the second is direct applications covering a wide temperature range and a variety of purposes (Figure 6). Most common is the use of water and geothermal energy for heating, balneotherapy, and recreation.

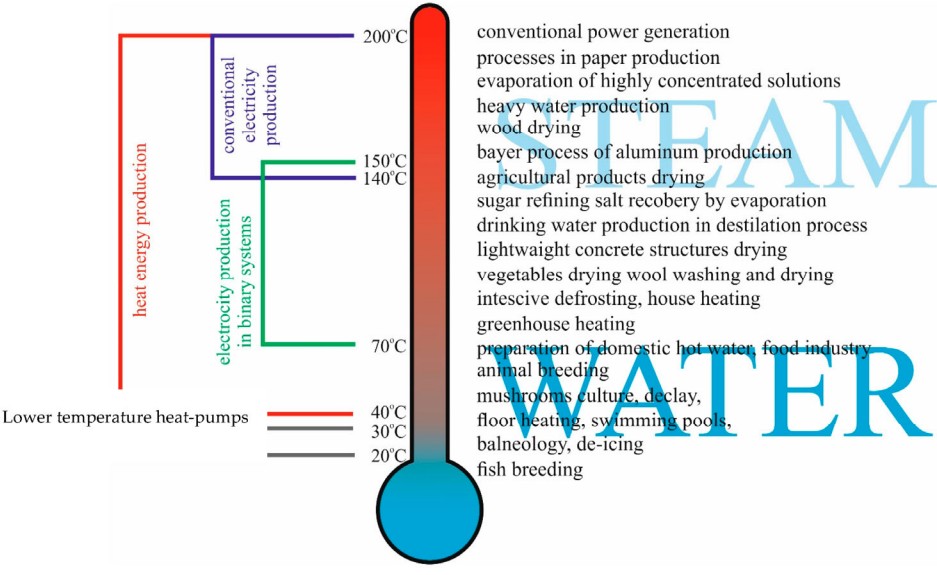

**Figure 6.** The possibilities of using geothermal waters depending on temperature (according to Lindal [30], with modifications). The green rectangle indicates the possibilities of using water for heating and cooling with the use of heat pumps.

The geothermal fluid temperatures required to utilize heat directly are lower than the temperatures needed for the economical generation of electricity, which makes using these resources more common. Heating rooms with geothermal energy is the key direct application, especially on an individual level. It should be noted that geothermal energy can also be used to cool rooms. Geothermal electricity generation has been possible since 1904. The first type of geothermal power plant was a dry steam plant, which relied upon a vapor-phase geofluid [31].

One of the simplest geothermal applications is pond or pool heating, as they typically use geothermal water directly to meet the required heat demand [32]. A particularly attractive way of using geothermal waters is agribusiness applications associated with low temperatures necessary for their utilization, made possible due to the prevalence of resources suitable for this purpose. The use of waste heat and, above all, the cascade use of geothermal energy, which increases the efficiency of geothermal projects, also provide

excellent opportunities. A number of agribusiness applications can be considered, including the following: heating greenhouses, aquaculture, and animal husbandry; the heating and irrigation of soil; the heating and cooling of mushroom crops; etc. The use of geothermal energy to breed catfish, shrimp, tilapia, eels, and tropical fish yields crops faster than the use of conventional solar heating. The utilization of geothermal heat allows for better control of the temperature of a fish pond, thus optimizing the growth of the fish. The most important factors to consider are water quality and disease. If geothermal water is used directly, the concentrations of dissolved heavy metals, fluorides, chlorides, arsenic, and boron should be taken into account and, if necessary, isolated with heat exchangers. Livestock facilities can stimulate the growth of farm animals through controlled heating and cooling. In confined spaces, juvenile offspring mortality can be reduced, growth rate and litter size can be increased, diseases can be controlled, waste management and collection can be made easier, and, in most cases, product quality can also be improved. In addition, geothermal fluids can be used to clean, sanitize, and drain animal shelters and waste, as well as to support biogas production from waste [32].

The beneficial effect of geothermal waters on the human body has been known and capitalized upon for many years with the use of therapeutic or recreational baths. Chemical and physical factors act on the human body during bathing. Unlike therapeutic baths, recreational baths cannot be highly stimulating due to the temperature (it cannot be either too low or too high) and due to mineralization (the salinity of the world ocean, equal to $35 \text{ g/dm}^3$, is assumed as the maximum value). The minimum capacity of geothermal water from the intake provided for one recreational pool should be from 3 to 5 $\text{m}^3/\text{h}$ [33]. The main physical parameter of water affecting the body during bathing is its temperature [22]. Depending on the temperature of the water, it is used for hot or cold baths. As part of therapy involving the use of low-temperature waters (i.e., with the temperature equal to 20–27 °C), the narrowing of blood vessels, increases in blood pressure, increases in muscle tone, and increases in metabolism can be expected. On the other hand, thanks to bathing in waters with a temperature ranging from 38 to 40 °C, the effects of vasodilation, lowered blood pressure, an increase in the efficiency of the circulatory system and organs such as the lungs or kidneys, and improvements in blood supply to tissues can be achieved.

Nevertheless, the most significant are the chemical components that exert their specific effects when penetrating the skin. Mineral components dissolved in thermal waters, mainly magnesium, calcium, and sodium chlorides, depending on their concentration, create the so-called salt mantle. This process causes the expansion of blood vessels and thus improves the condition and blood supply to the outer layers of the skin. In addition, it affects the endings of motor and sensory nerves, reducing their excitability and the feeling of pain [34]. There are also noticeable benefits resulting from the presence of certain minerals in the waters, such as [22] iodide anions (they have bactericidal and anti-inflammatory effects on the skin surface and heal thyroid diseases), sulfides (they have an antioxidant effect and beneficially affect the regeneration of joint cartilage), and radon (which is anti-inflammatory and acts as a pain reliever). The minerals contained in geothermal water can provide the skin with the micro and macro elements it needs, which improves the rate of skin regeneration. Additionally, the water has soothing and anti-inflammatory effects, causing cosmetics based on thermal waters to gain more and more popularity.

According to Polish law [35], based on the chemical composition of water and its properties, groundwaters can be classified as the following:

(1) mineral waters—waters containing at least 1000 mg of dissolved components per 1 $\text{dm}^3$, including sodium, calcium, and magnesium chlorides, as well as sulphates and bicarbonates occurring in individual waters in various quantitative ratios. The characteristics of mineral water include the following: the percentage content of total solids dissolved in a given water, the names of anions and cations whose quantitative share in milligram equivalents (meq) in 1 $\text{dm}^3$ of water exceeds 20% in the order of decreasing concentrations;

(2)　specific (slightly mineralized) waters—waters containing less than 1000 mg of dissolved solids per 1 $dm^3$, including one or more specific medicinal components in the concentrations specified below or higher:

(a)　1 mg of iodides—iodide water;

(b)　1 mg of sulfides or other sulfur compounds (II)—sulfide water;

(c)　2 mg of fluorides—fluoride water;

(d)　10 mg of iron (II)—iron water;

(e)　70 mg of metasilicic acid—silicon water;

(f)　1000 mg of free carbon dioxide—acidulous water;

(g)　250—999 mg of free carbon dioxide—carbonic acid water or

(h)　showing a temperature of at least 20 °C at the outflow from the intake—thermal water;

(i)　showing radioactive activity of at least 74 $Bg/dm^3$—radon water.

The characteristics of a specific water lists the typical components it contains in the order of decreasing concentrations;

(3)　specific mineral waters—mineral waters containing one or more specific components listed in point 2.

The characteristics of specific mineral water includes the percentage content of dissolved minerals, the names of anions and cations whose quantitative share in milligram equivalents (meq) per 1 $dm^3$ of water exceeds 20%, and the name associated with the specific components typical of a given water, in the order of decreasing concentrations.

One of the non-energetic aspects of geothermal water management is also its use for food purposes. Geothermal water can be used for the production of drinkable curing agents, as well as for bottling and food production, depending on its chemical composition. In the case of the occurrence of saline groundwater, desalination has become a frequently used method of obtaining water intended for consumption and for economic purposes. The water treatment market is also driven by the increasing demand for water, caused by the growing population and changes in people's lifestyles [36,37].

*4.2. The Social Benefits of Using Geothermal Waters*

There are a number of factors that decide the profitability of a geothermal project. Of course, the most important are the hydrogeothermal conditions in a given region [4]. This element is unchangeable and cannot be influenced. However, there are examples of cases in which investments have not been successful despite favorable geothermal conditions. Effective resource management also depends on the existence of the energy recipient and the proper design of the geothermal installation. The competitive position of a geothermal energy carrier, as well as the availability and cost of capital allocated to geothermal investments, will be influenced by the costs of heat production using conventional methods, as well as the pro-ecological policy of the state and the amount of funds allocated to scientific research and support for geothermal investments. These aspects change with time and vary widely across European countries.

One of the essential elements of achieving success in the use of geothermal waters is the social awareness of the inhabitants of the commune—both policymakers and all citizens and visitors [38,39]. The results of the research conducted by Smętkiewicz [38,39] in the Uniejów health resort, which uses geothermal waters as a natural resource, clearly show that, according to the health resort's residents and visitors, geothermal energy plays a key role in the favorable socio-economic changes in the town (Figure 7). The use of geothermal waters primarily contributes to increasing the attractiveness of the town and even the entire region to tourists, improving the status of the local labor market and economic development. All these elements generally translate into improving the quality of life of the local community.

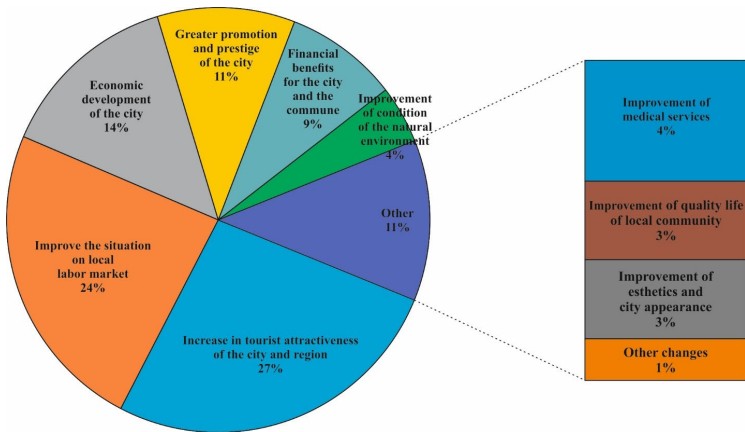

**Figure 7.** Opinions of residents of the town and municipality of Uniejów on positive changes resulting from their community possessing a health resort (based on Smętkiewicz [39]).

Taking into account the abovementioned research results, the benefits associated with the use of geothermal energy also include security related to the stability of supplies and the availability of an energy source such as groundwater, regardless of the time of day or year, convenience of use without the need to purchase and install a heating furnace, and predictability related to the prices associated with the use of geothermal energy (Figure 8). These factors, along with the favorable environmental impact associated with reducing the emission of harmful substances into the atmosphere, contribute to the development of an entire region.

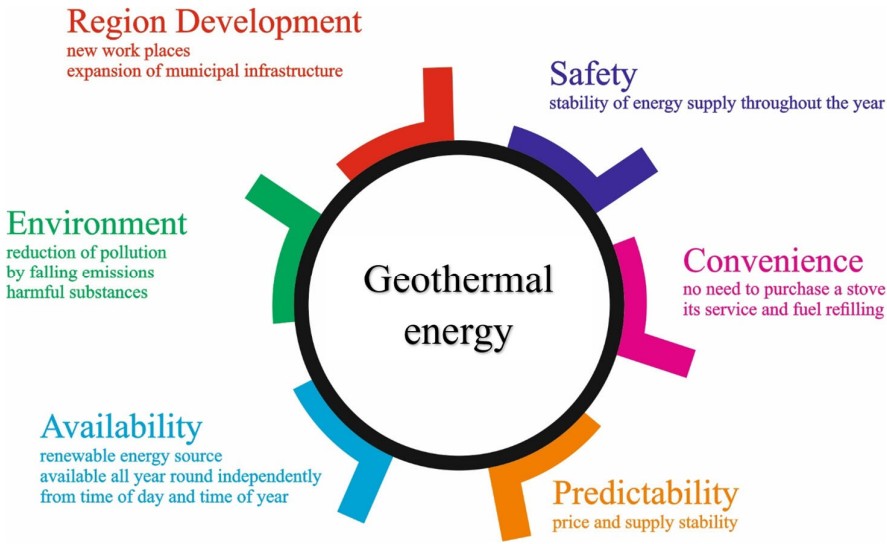

**Figure 8.** The main benefits of using geothermal energy.

In addition to energy applications that bring significant environmental benefits, geothermal waters can be used for recreation, balneotherapy, and cosmetology. These are the applications that attract tourists or patients to visit a region and enjoy the benefits of geothermal waters. From the point of view of tourism development, the use of geothermal resources has the following main functions [40]: spa and recreation, balneotherapy, heating and electricity generation (if possible, depending on their temperature (Figure 9), chemical composition, and the presence of mud). The development of tourism influences the economic development of a region. Accommodation facilities are being built or modernized, catering services are being developed, and areas for recreation are being created. The quality of life of the inhabitants is improved through the development of urban infrastructure and the emergence of new investments in the city. The energy application of geothermal

resources affects air quality, as the amount of pollutants emitted is the result of burning solid fuels, which is currently a huge problem in many towns (and even tourist resorts) across Poland. The results of the research presented in the literature confirm the existence of a number of benefits for the community related to the extensive use of the local geothermal resources [41,42]. The significant socio-economic and tourism development of the region can be observed.

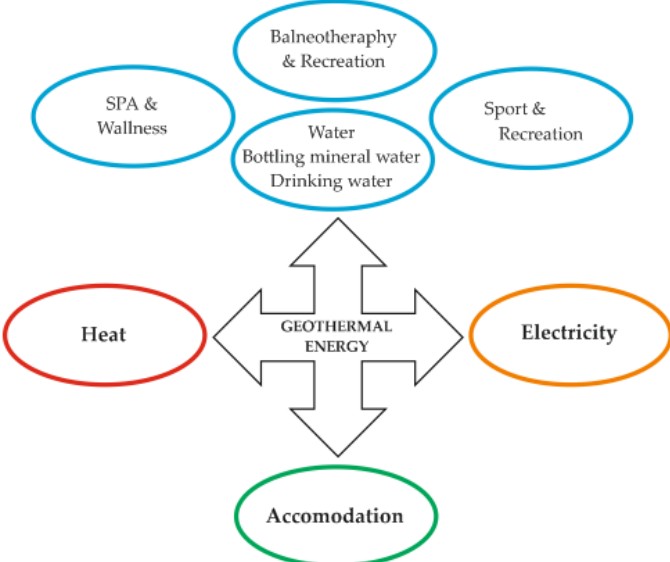

**Figure 9.** Main functions related to possibilities of using geothermal waters from the point of view of tourism development (adapted from Ref. [40]).

### 4.3. Predicting Geothermal Energy Production for the Commerce and Public Service Sectors in Poland until 2030

Taking into account the degree of the use of geothermal energy in Uniejów and Poddębice, as well as the possibilities of introducing the use of geothermal energy in regions such as Malanów, it can be assumed that this RES sector will develop in the coming years. If this increase is similar to the increase that took place between 2017 and 2021 (according to data from the Central Statistical Office), we can expect to achieve a level of approximately 1600 TJ geothermal energy in 2030 (Figure 10). This prediction was made assuming that the geothermal energy sector will experience the stable growth observed in the previous years in Poland.

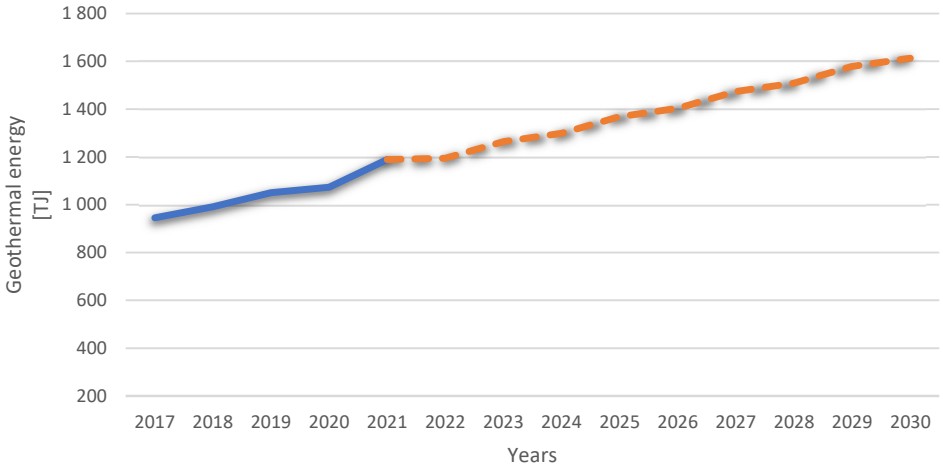

**Figure 10.** Prediction (orange dotted line) of geothermal energy production for the commerce and public service sectors in Poland until 2030 (based on Central Statistical Office data—blue line).

It should be emphasized that the rate of increase in the share of geothermal energy in the entire RES depends on many factors. The most important of these factors are the economic challenges associated with drilling and reservoir development. Also, financial issues for geothermal power projects include the high upfront costs and the long return-on-investment timeframes, which may significantly delay the development of this RES sector.

## 5. Conclusions

The use of geothermal energy resources, in addition to the undoubted benefits related to the possibility of obtaining clean energy, brings many social benefits related to the development of sectors such as the tourism, cosmetology, food production, and many other sectors, as exemplified by cities such as Uniejów or Poddębice. Given the dynamic development of cities with geothermal heating plants or centers that use groundwater, geothermal investments constitute a new development perspective, which is especially important for regions that are poor in mineral resources.

One of the most promising regions in Poland for the use of water resources and geothermal energy is the region located in central Poland within the Mogilno–Łódź Trough (Polish Lowlands). Currently, two of the six geothermal heating plants in Poland are successfully operating within this region (in Uniejów and Poddębice). They are excellent examples of the multifaceted applications of geothermal waters, which range from using geothermal waters for heating purposes, balneotherapy, recreation, and cosmetology to, for example, heating the turf of a football field, or pickling cucumbers based on using geothermal water. Such a vast array of applications, in addition to yielding typically economic benefits, also yields social benefits. These regions have become more attractive, and the use of geothermal waters is seen as a unique selling point for the local area. The indigenous community reaps great benefits, which is reflected in the development of the region and improvements in the quality of the environment resulting from reducing solid fuel combustion.

Taking into account the potential of the Mogilno–Łódź Trough region, there are still areas within its borders where further possibilities for the management of geothermal resources are suspected or have already been identified. In light of the research conducted in recent years, promising regions include, among others, cities such as Konin or Turek, where wells have now been made to confirm the possibility of using groundwater in the Lower Jurassic reservoir. In the present article, attention was drawn to the possibility of also using geothermal energy in the area of Malanów, where the waters of the Lower Jurassic deposit that can be managed exceed a temperature of 80 °C at a capacity of up to 120 m$^3$/h, and the potential of these waters could be extensively exploited in this area. It should therefore be emphasized that in the case of a new investment in the Malanów region, the potential recipients may turn out to be the key aspect in determining the profitability of the project, as the area's relatively large distance from larger cities such as Kalisz may, on the other hand, translate into the inability to implement geothermal investments in this area despite favorable geothermal conditions. Nevertheless, the example of Malanów proves that the geothermal potential of the Mogilno–Łódź Trough remains undiscovered and may contribute to an increase in the use of groundwater throughout the country in the coming years. Additionally, given the dynamic development of the regions currently using geothermal water resources, future projects will certainly bring measurable results, both for the local community and the environment, by reducing pollution.

**Author Contributions:** Conceptualization, M.S.; Methodology, A.W.-P.; Validation, A.S.; Investigation, T.M.; Data curation, T.M.; Writing—original draft, A.W.-P.; Writing—review & editing, A.S.; Visualization, A.W.-P.; Supervision, M.S. All authors have read and agreed to the published version of the manuscript.

**Funding:** The paper has been prepared under the AGH- AGH University of Krakow statutory research grants No. AGH 16.16.140.315./05.

**Institutional Review Board Statement:** Not applicable.

**Informed Consent Statement:** Not applicable.

**Data Availability Statement:** The data presented in this study are available on request from the corresponding author.

**Conflicts of Interest:** The authors have no conflict of interest that are relevant to the content of this article to declare.

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
