# Peer review of "A New Approach to the Development of Geothermal Water Utilization in the Context of Identifying and Meeting the Social Needs of Local Communities: A Case Study from the Mogilno–Łódź Trough, Central Poland"

_sustainability, doi:10.3390/su16010037_

Round 1

Reviewer 1 Report

Comments and Suggestions for Authors

Dear authors,

this is an interesting paper with focus of the research aimed at providing the results for better understanding geothermal waters utilization. Studying this paper I consider that the manuscript is a new attempt about a subject. It is based on many other studies about geothermal water utilization  performed before.

Title is too long,

The topic is interesting.

There is no metod described, it seems that paper has no methodology/methods of research. Authors need to add information when the research was done and how (interview, focus groups, questionnaire etc), also to explain research sample.

Tables and figures are clearly presented.

I suggest authors to expand chapter "methodology". After improving the methodology, results and discussion will also need to be improved.

I found that the literature review section and methodology are the main lacks of the paper. I suggest authors to expand paper with chapter "Literature review". Chapter “Conclusion” is too theoretic, there is generally description predominance.

Beside that fact, I think that this paper could still be a good contribution to theory and practice, after adopting suggestions and corrections.

Author Response

Thank you very much for your valuable comments. We have tried to respond to all the  submitted suggestions. 

Reviewer 2 Report

Comments and Suggestions for Authors

The article A new approach to the development of geothermal water utilization in the context of identifying and meeting the social needs of local communities: case study from Mogilno-Lódz Trough, Central Poland addresses the issue of the development of geothermal water utilization from the perspective of the social needs of local communities in Poland . As a source of renewable energy, geothermal energy is in last place, after biogas, hydropower, municipal waste with a share of less than 1%. The authors provide information about the potential of geothermal energy in Poland, especially in the Polish Lowlands. After the detailed description of the methods of use in the present and in the future, the authors highlight the social benefits of the use of geothermal water.

For the future development of the use of geothermal water, it is necessary to outline the whole picture, not just the benefits. The reasons why geothermal energy has not been exploited more widely are the economic challenges associated with drilling and reservoir development. Virtually all geothermal drilling technology initiatives come from the oil and gas industry. Unlike the oil and gas industries, however, rapid returns on investment are much lower in geothermal power production.  Chronic financial issues for geothermal power projects include the high upfront costs and the long return-on-investment timeframe. These factors could discourage private investors. Unlike solar and wind, finding appropriate locations for geothermal requires expensive geological exploration, and many geothermal systems are completely invisible from the surface. In order to accurately quantify the potential of the field, developers must invest heavily in exploration technologies.

The most significant impacts include gaseous emissions, water use and consumption issues, land use issues, drilling risks, seismic activity, land subsidence, and reduction of thermal features.

Author Response

Thank you very much for your valuable comments. We have tried to respond to all the suggestions submitted.

Reviewer 3 Report

Comments and Suggestions for Authors

Points that need to be considered:

  1. A global comment:

The text requires careful editing in terms of style/grammar and sentence structure. Sometimes the sentences are very long and complex, which makes reading difficult. In some cases, the sentences are grammatically incorrect. Example:

"The energy application of geothermal resources affects air quality as the amount of pollutants emitted as a result of burning solid fuels, which is currently a huge problem in many towns (and even tourist resorts) across Poland, is reduced."

It is not known what the phrase "is reduced" refers to.

  1. Line 188:

It is "utility water".

It should be: domestic hot water

  1. Figure 6.

There is "lower heat-pumps".

I propose to change it into: low temperature heat-pumps

  1. Line 387:

It is "According to Polish law [32], (...)".

Please check if the numbering of the literature is correct in the entire manuscript.

Eg. particularly in this case it needs to be checked if it shouldn't be [40] (Polish legal act), instead of [32] (a foreign paper).

  1. Line 464:

There is an opening bracket, but no closing bracket.

Comments on the Quality of English Language

The text requires careful editing in terms of style/grammar and sentence structure. Sometimes the sentences are very long and complex, which makes reading difficult. In some cases, the sentences are grammatically incorrect. Example:

"The energy application of geothermal resources affects air quality as the amount of pollutants emitted as a result of burning solid fuels, which is currently a huge problem in many towns (and even tourist resorts) across Poland, is reduced."

It is not known what the phrase "is reduced" refers to.

Author Response

(The authors gave the same response as above.)

Reviewer 4 Report

Comments and Suggestions for Authors

lines 25-28 please be more specific

lines 182-193 please explain which are your contributions to the heating system

lines 263-267 please give explanations of the modelling method or algorithm

a logic diagram of the proposed model/algorithm ,vcoild be a good option for the presentation

table 2 - please explain 

table 3-please explain 

lines 481 - 487 please give explanations of the prediction method proposed  

conclusions -  please explain your scientific contributions about  - your poposed scientific model - primary information  data base  - algorithm of processing  the information  - results data base 

Comments on the Quality of English Language

moderate english improvmet 

Author Response

(The authors gave the same response as above.)

Round 2

Reviewer 1 Report

Comments and Suggestions for Authors

Dear authors, thank you for accepting suggestions and made changes in some parts of manuscript,

Reviewer 3 Report

Comments and Suggestions for Authors

None.

Reviewer 4 Report

Comments and Suggestions for Authors

the paper is accepted 

Comments on the Quality of English Language

please check the english in details